# A Novel Privacy Paradigm for Improving Serial Data Privacy

**DOI:** 10.3390/s22072811

**Published:** 2022-04-06

**Authors:** Ayesha Shaukat, Adeel Anjum, Saif U. R. Malik, Munam Ali Shah, Carsten Maple

**Affiliations:** 1Department of Computer Science, COMSATS University Islamabad, Islamabad 45550, Pakistan; aysh2850@gmail.com (A.S.); mshah@comsats.edu.pk (M.A.S.); 2Institute of Information Technology, Quaid-e-Azam University Islamabad, Islamabad 15320, Pakistan; aanjum@qau.edu.pk; 3Cybernetica AS Tallinn, 12618 Tallinn, Estonia; saif.rehmanmalik@cyber.ee; 4Warwick Manufacturing Group (WMG), University of Warwick, Coventry CV4 7AL, UK

**Keywords:** preserving privacy, serial publication, multiple sensitive values, attribute disclosure attacks, transactional data

## Abstract

Protecting the privacy of individuals is of utmost concern in today’s society, as inscribed and governed by the prevailing privacy laws, such as GDPR. In serial data, bits of data are continuously released, but their combined effect may result in a privacy breach in the whole serial publication. Protecting serial data is crucial for preserving them from adversaries. Previous approaches provide privacy for relational data and serial data, but many loopholes exist when dealing with multiple sensitive values. We address these problems by introducing a novel privacy approach that limits the risk of privacy disclosure in republication and gives better privacy with much lower perturbation rates. Existing techniques provide a strong privacy guarantee against attacks on data privacy; however, in serial publication, the chances of attack still exist due to the continuous addition and deletion of data. In serial data, proper countermeasures for tackling attacks such as correlation attacks have not been taken, due to which serial publication is still at risk. Moreover, protecting privacy is a significant task due to the critical absence of sensitive values while dealing with multiple sensitive values. Due to this critical absence, signatures change in every release, which is a reason for attacks. In this paper, we introduce a novel approach in order to counter the composition attack and the transitive composition attack and we prove that the proposed approach is better than the existing state-of-the-art techniques. Our paper establishes the result with a systematic examination of the republication dilemma. Finally, we evaluate our work using benchmark datasets, and the results show the efficacy of the proposed technique.

## 1. Introduction

In today’s modern technologically connected society, the unending relationship between developing technologies and human beings induces individuals to share their data. Such data may include individual and personal confidential information—for example, electronic health records (EHRs), personal profiles, bank records, business data, etc. The PPDP initiative offers tools and approaches for meaningfully publishing such data while maintaining individual privacy. Recently, a huge amount of attention has been received from both the industry and academia, and many data privacy scenarios that use different types of data have been published [1]. Transactional data (or set-valued data) are one of the types of data where each record (i.e., a transaction) includes a set of items deduced from a universe or bunch of items [2]. The continuous arrival of data at a high speed (e.g., BigData) creates a data stream that grows rapidly and is potentially unbounded because intelligent computing devices and devices equipped with the Internet of Things (IoT) technology are very well liked [3]. Although a study on static transactional data also exists [1], due to the data-stream nature of transactional data, these approaches cannot be applied directly. Therefore, protecting transactional data in serial publication is challenging.

Due to the continuous addition and deletion of transactional data, the existing approaches can not preserve the privacy of the data. Many research institutes have provided the data for the analysis of the data however, this results in disclosure of some sensitive attributes [4]. The catastrophic results are due to an increase in risk factors, which has a significant impact on financial matters [5]. Different privacy-preserving approaches are used by the organizations to publish their data. Names and security numbers are often not released, but due to linkage, sometimes, the data are violated [6]. Datasets have faced multiple releases in different orders that are needed by users [7]. When a person exists in numerous datasets, an attacker can easily leak data, although they are preserved in all unrelated publications [8]. This type of attack is called correlation [9]. Hackers attack the confidential data of individuals by using independent datasets, often through composition attacks. A technique called cell generalization is proposed as a solution. Some attacks, such as composition and transitive composition attacks, are discussed in detail in [10], and they show how data are serially preserved. However, there are still several assaults on which we shall focus in our study. Attacks on Sanony have been discovered due to the crucial lack of sensitive values and the link between sensitive and non-sensitive attributes. We have a thorough examination of privacy and security theory in a global bag, a concept that was introduced in [10]. By combining all publicly accessible information, we explain how an adversary can recreate microdata. The formulas and our technique illustrate why previous generalization rules failed and contribute to the creation of the so-called m−invariance in the global bag. We demonstrate that the chances of exposure are effectively lower due to the current concept. This scheme is applied to real-time base datasets, e.g., health records. Table 1 shows notations used in the paper.

### 1.1. Motivation

Privacy protection is a prominent issue for study, whether it concerns a person or a company. One might lose a lot of money if one makes even a tiny error. Therefore, we propose an approach in order to pay close attention to the handling dynamic data. Because attackers primarily employ quasi-identifiers (QIs) or a non-private section of a transaction to gain information, we must alter the QI values in such a manner that they are rendered worthless to them. Due to their spying nature, attackers frequently maintain an eye on their target. When people smoke in public locations, it demonstrates their addiction or increases their risk of lung cancer. Therefore, our aim is to solve these problems, since the adversary is busy guessing or gaining information from the victim’s actions.

The authors of [10] suggested a privacy approach for serial data posting that deals with a large number of sensitive variables. They suggested the notion of a “global bag” for dealing with many sensitive variables. Due to the critical absence and the continual addition of data in subsequent versions, the global bag is vulnerable to attribute disclosure attacks. Therefore, our aim is to provide an effective serial-data-publishing privacy model in order to accommodate the global bag’s numerous sensitive values and important absence. Many generic words may be used to justify a violation. Diabetes may be diagnosed if someone begins to lose weight without trying, has dry skin, or refuses to eat sweets or desserts. Our objective is to enhance serial data privacy to defend against intrusions more than in past ways.

### 1.2. Contribution

It is a difficult undertaking to keep serial data private. There is much literature on the publication of serial data [10,11]. Given the fascinating nature of such data, serial data publishing remains an unresolved challenge. Serial data are vulnerable to various attacks due to their dynamic nature (updated data are released regularly). When numerous sensitive values are present, the problem becomes more complicated. However, as shown in [10], if a record is removed from the global bag, an attack on the following release may be carried out if there is a significant absence of that record in the global bag, thus disrupting individual privacy. The goal of this work is to protect the global bag in serial publishing. Another contribution of this work is that there may be the addition of records, since the data are serial. In this study, we will emphasize the significance of guarding against data addition. When a record is added to a release, it is not included in the global bag of prior releases, leading to increased data vulnerability. We will theoretically and practically verify our statements by analyzing these concerns in depth in the following sections. Continuous releases of microdata are shown in Table 2, Table 3 and Table 4. Anonymized prisoner health records are shown in Table 5, Table 6 and Table 7. If $t_{5}^{*}$ is erased after the first release in Table 5, it is not included in the global bag of the second release, which allows the $t_{5}^{*}$ data to be readily breached, as shown in Table 6.

Furthermore, we can see in Table 6 that a $t_{7}^{*}$ record has been added, resulting in diabetes being added to the global bag, though it was not there in the prior year’s record. As a result, our contribution is to address the difficulties mentioned earlier. Now, we will look at another factor that contributes to the rise in data breaches. Each preceding record is released in serial data, allowing an opponent to steal data from it. In this case, non-sensitive portion is related to the disclosure of prisoners’ medical records which is connected with the nature of their offenses. The majority of information is gathered from prior knowledge. Each corpus is serially stored in the transactional data, but the total publication may be jeopardized due to correlation. Although several ways to deal with correlation attacks have been offered in the past, the uniqueness of our contribution is that we overcome correlation attacks when dealing with many sensitive variables in serial publishing. Due to the correspondence between sensitive and non-sensitive elements, we have provided a technique for coping with these assaults on transactional data. Assume that the opponent is searching for record $t_{2}^{*}$ in Table 5. If the opponent learns that $t_{2}^{*}$ has been arrested for a law-breaking crime of smoking in public areas where smoking is prohibited, this belongs to the first equivalence class in each year. If they are addicted to smoking, there is a high probability that $t_{2}^{*}$ will get lung cancer, as indicated in Table 5. It can be noted that it is logged in each year and the opponent can easily discover the sickness of $t_{2}^{*}$ due to the link between the sensitive and non-sensitive attributes. If $t_{2}^{*}$ smokes, their chances of developing lung cancer increase. Cluster 1 contains only the history of $t_{2}^{*}$ in Table 6 owing to the deletion of the record of $t_{1}^{*}$. Even with counterfeiting, the method provided in [10] for coping with composition attacks, $t_{2}^{*}$ is more likely to be subjected to a correlation attack. Furthermore, our article reduces risk by protecting serial data against previously known attacks, such as composition and transitive composition attacks, on serial data, as discussed in Section 4.2, and against recently discovered techniques, such as correlation and critical absence attacks, as discussed in Section 4.3. This research is significant, as this technique successfully minimizes the danger of disclosure and protects individual privacy, which is of great importance in today’s world. For calculating m-invariant publishable relations, we offer a feasible approach. We safeguard the global bag by introducing the number of counterfeit tuples, and we categorize non-sensitive characteristics to defend against correlation attacks. Our method provides a higher assurance of privacy. Our contributions are illustrated here:Protecting the global bag in serial publication.Dealing with correlation attacks.Preserving individual privacy.Minimizing risk factors.

Our contributions, as mentioned above, are critical in serial publishing. Because individual privacy is a top concern, we have highlighted our efforts in this section. We show that our method has a lower risk than that of prior methods. To reduce risk, a viable and resilient relative privacy paradigm is provided. The rest of the sections are organized as follows. Section 2 reviews the previous literature. Section 3 explains the basic concepts needed to understand the nature of serial data. Section 4 presents the proposed methodology, along with algorithms and mathematical formulations. Section 5 experimentally explains the effectiveness of our approach. Section 6 provides the conclusions and gives future directions for this work.

## 2. Literature Review

The literature contains methodologies for protecting data for one-time release. Note that the literature does not focus on providing the security for republication of micro-data. In 2007, a technique called m-invariance was proposed, which deals with disclosure issues in republication [12]. Among privacy protection mechanisms, k-anonymity is one of the most essential anonymization techniques, but generalizing it has many drawbacks [13]. In serially published data, a record is added or deleted with consequences for time and privacy disclosure. Consider an example of serial data—e.g., prisoner health data that are released every year, as described in [1,14]. Anatomy is another technique for publishing data without generalization [15]. K-anonymity does not work well for a diverse set of sensitive attributes. Hence, l-diversity was proposed to resolve these issues [16]. Further analysis revealed that t-closeness degrades the usefulness of data. As a result, a more compact model (n,t) closeness was proposed [17]. In the past, privacy was provided only for static data, but in reality, datasets are dynamic, so we used anatomy in the place of generalization and suppression [15].

Among the various privacy approaches, differential privacy has received attention, but it results in loss in utility when anonymizing data [18]. Microdata publishing is essential for scientific research, analytical techniques, and data mining. Techniques such as SLOMS and SLAMSA are, indeed, available for the anonymization of several sensitive features, but they have limitations regarding privacy and data value. One work proposed a fast (p, k)-Angelization method, but it did not work with multiple sensitive values [19,20]. Even though the newly presented approach of (p, k)-Angelization provides a unique solution, researchers explored the risk of privacy leaks through the correlation of multiple sensitive attributes across linkable sensitive buckets and called this a fingerprint correlation assault. Another paper proposed a privacy approach that converted one-to-many communication [21]. Most privacy schemes work on relational data, but not on transactional data. This paper provided a technique called anony for transactional data. Techniques such as generalization, partitioning, and perturbation make terms more general [22,23,24], break links [15,25] in data, and add noise [26,27,28], respectively. A skewness attack is handled through t-closeness and differential privacy, but has many disadvantages. Interested readers can read [11]. Differential privacy bears many unsolved challenges [26], for example, when two or three datasets differ in only one record. Thus, a transaction should be independent, which is not possible in serial data [29]. For this, the concepts of m-invariance and counterfeit generalization were introduced, where counterfeits were added to protect from breaches in dynamic data [12].

M-invariance only deals with external updates in which data are added or deleted once, and it is not concerned with internal updates. To deal with the drawback of m-invariance, a technique called *t*-safety was proposed, which deals with external and internal updates [30]. Publishers are well aware of overlapping records and protect them from composition attacks, thus providing maximum privacy [31,32]. In the collaborative publication of multiple data, a member can hold only a specific portion of the dataset, and all members jointly publish overall datasets [33,34,35,36,37]. Cell generalization further improves the utility [38]. One hybrid approach was proposed in [39] to reduce noise by using generalization to protect data from attacks such as identity and attribute disclosures. The MCMC methodology was proposed to learn population parameters [40]. Another paper proposed an anonymization technique called disassociation, which preserves original data and protects data by hiding [41]. Data privacy protection for one to M datasets (MSAs) is exciting and challenging. The k-anonymity model is often used to investigate privacy exposures, but it is only useful for identity disclosure. Using adversarial situations, this study paid attention to the behavior of an enemy that disclosed identities and attributes. Thus, satisfying the “p+-sensitive k-anonymity model” is insufficient [42].

Anonymization strategies enable data collection while retaining privacy. Anonymization models, however, involve harsh or unreasonable assumptions regarding performance evaluations. One study discovered and defined the “Leader Collusion Attack (LCA)”; they proposed a dynamic data collection strategy based on k-anonymity in order to decrease LCAs [43]. Another study proposed the Reversible Data Transform (RDT) algorithm, which maintains privacy [44]. Privacy was further improved in sequential releases with full dynamic settings in which different releases had different groups of tuples and attributes [45]. IncTDPART top-down portioning was also proposed for incremental updates [46]. Many disclosure issues exist when dealing with serial data, as shown in Figure 1. Our approach helps deal with attribute disclosure issues in serial data.

## 3. Preliminaries

In this section, we design some fundamental principles that shape the basis of republication, which leads to privacy disclosure. A list of symbols is shown in Table 1.

**Definition** **1**(Nature of serial data)**.** *As serial data contain corpora, E is dynamic in nature, and data are published in an anonymized version at frequent timestamps $T_{1}^{*},T_{2}^{*}\ldots \ldots T^{*}\left(n\right)$; then, these timestamps are somehow connected. Due to republication, the chances of a data breach increase. In one timestamp, $T_{1}^{*}=t_{1}^{*},t_{2}^{*},t_{3}^{*}\ldots t^{*}\left(n\right)$, where n > 1, and in the second timestamp, $T_{2}=t_{3}^{*},t_{4}^{*}\ldots t^{*}\left(n\right)$. Then, due to the republication of $t_{3}^{*}$, many types of attacks can easily be performed.*

**Definition** **2**(Republication)**.** *Let transactions $t_{1}^{*},t_{2}^{*},t_{3}^{*}\ldots \ldots t^{*}\left(n\right)$ be published by a publisher such that all transactions $t^{*}\left(n\right)\in T_{1}^{*}$. Then, another set of transactions are published in $T_{2}^{*}$, which contains some transactions from Release 1, which is known as republication. The objective is to calculate the risk in dynamic data from different releases due to republication.*

**Definition** **3**(Background knowledge)**.** *At time n, the adversarial knowledge about the victim is information that may include some background knowledge B(n), which includes data published in previous releases and knowledge about the non-sensitive portion ${\bar{T}}_{s}$ of a transaction.*

**Definition** **4**(Tuple union)**.** *At time n≤1, the tuple union contains all tuples at time i,j…n. Each tuple $t^{*}\in U^{*}\left(n\right)$ is associated with lifespan (i,j). Note that if a tuple appears several times in t(j), it is included only once in $U^{*}\left(n\right)$*
(1)$$U^{*}\left(n\right)=\overset{n}{\underset{i=1}{⋃}}t\left(i,j\ldots n\right)$$

**Definition** **5**(Privacy breach)**.** *If an attacker finds out the sensitive value of any transaction $t^{*}\in U^{*}\left(n\right)$ by utilizing $t_{1}^{*},t_{2}^{*}\ldots t^{*}\left(n\right)$ and background knowledge B(n), it results in a privacy breach for that individual.*

**Example** **1.**
*Let us take an example where an attacker has the prior information that a victim has been arrested due to the law-breaking offense of smoking. He might have lung cancer; here, chances of a data breach exist.*


**Definition** **6**(Surjective functions)**.** *If A and B are two sets of transactions, one containing the union of transactions of all releases, represented as $U^{*}\left(n\right)$, and the other containing the background knowledge B(n) of the adversary, then, surjective functions and the ”onto” function are defined, as for every element in B(n) there exists an element in $U^{*}\left(n\right)$, which is also known as a rebuilding surjective function.*

**Definition** **7**(Reasonable surjective function)**.** *If row b ∈ B(n) carries the same sensitive value as that in $U^{*}\left(n\right)$, then it is known as a reasonable surjective function. A function may map one or more elements of $U^{*}\left(n\right)$ to the same element of B(n).*

**Definition** **8**(Global bag)**.** *Let $t_{1}^{*},t_{2}^{*}\ldots t^{*}\left(n\right)$ be a set of transactions containing multiple sensitive values at time i, and all of the second sensitive values are included in a special bag known as the global bag $T_{g}$. The global bag is globally attached to the whole corpus $T^{*}$ for protection from data breaches. For example, the global bag in Table 5 contains herpes and COVID-19.*

**Definition** **9**(Global sensitive set (GSS))**.** *Let $t^{*}$ belong to a GSS at time j. The global sensitive set contains all sensitive values of the global bag from i,j…n. If this set contains herpes and COVID-19 at timestamp i and COVID-19 and diabetes at timestamp j, then it is proved by a lemma that a transaction’s sensitive value can be compromised. Hence, it is proven that if $n_{breach}\left(t\right)$ = $n_{total}$, an attack can be easily performed on the global bag. Sensitive data include medical data, personal profiles, bank data, business data, etc.*

**Definition** **10**(Critical absence)**.** *Let $t_{1}^{*}$ and $t_{2}^{*}$ be two transactions containing sensitive values, and $t_{2}^{*}$ has one sensitive value in the global bag Cg; then, if $t_{2}^{*}$ is deleted, its absence may cause the removal of that sensitive value from the global bag. Then, the critical absence holds in the global bag for the next release, thus leading to an attack because the sensitive value of $t_{2}^{*}$ has been removed from the time (i…j). Our data can be serially preserved if the risk is less than 1/m. If the above condition is satisfied, it is shown that our data are serially preserved.*

## 4. Proposed Methodology

This section presents our proposed methodology, Kanony, for tackling different kinds of attacks on serial data, as shown in Figure 2. It provides security for the overall data, but the main focus is on the security of the global bag, which is neglected by Sanony [10]. Our solution’s foundation is m-invariance [12], whose fulfillment means that classified material is well protected through republication.

### 4.1. Anonymization

Firstly, we anonymize our data in two steps, as in [11]. Given two corpora $T_{1}$ and $T_{2}$, by applying segregation ($T_{s},{\bar{T}}_{s}$) and sanitization ($T_{s},{\bar{T}}_{s},T_{g}$) to all clusters, we obtain $C^{*}$ anonymized clusters. These anonymization techniques must be used to accomplish our suggested strategy. We must first develop a global bag, since we are working on global bag security. We first separate private and non-private segments, and then the private segments are further split into cluster private and global private segments. We can protect ourselves against composition attacks, transitive composition attacks, our newly discovered correlation attack, and critical absence, as well as additional concerns, with respect to the global bag by performing anonymization and applying overall perturbation in the private segments and categorization in the non-private segments.

#### 4.1.1. Segregation

In this part, vertical partitioning occurs by dividing the sensitive and non-sensitive portions ($T_{s},{\bar{T}}_{s}$). As we know, all transactions have more than one sensitive value, and due to this, many attacks can be performed, such as minimality attacks. So, we need to divide this sensitive portion in the sanitization step by separating two sensitive values to protect against these attacks.

#### 4.1.2. Sanitization

Sanitization further partitions all private terms into two parts according to their requirements. These two parts are the cluster private and global private segments ($T_{s},T_{g}$). For all clusters, these steps are repeated because serial data are dynamic, and this technique is applied to all clusters to get a fully anonymized corpus $T^{*}$. After completing these steps, we are still vulnerable to composition and transitive composition attacks, which may be mitigated by using perturbation in the cluster private segment.

### 4.2. Backward Perturbation (BP) and Forward Perturbation (FP)

We make two subsegments of the private segment, namely, the global private segment and cluster private segment. We are working with serial data in which records are continuously added and deleted. Various attacks arise after segmentation, including composition and transitive composition attacks. To overcome these attacks, backward and forward perturbation is introduced, which involves the use of counterfeits in the cluster private segment. Backward perturbation adds counterfeits in the cluster private segment to reduce “composition assaults” resulting from transactions being connected to a previously disclosed corpus. Because of these counterfeits, the overlap is reduced by altering the cover $\Omega (C^{o})$, which is a group of overlaps between two corpora. Through backward perturbation, overlaps with previously published corpora can be reduced. Next, FP is applied on the derived clusters by adding further counterfeits to protect against transitive composition attacks. The derived clusters are clusters formed after the addition and deletion of records in subsequent releases. We are aware that records are constantly being added and deleted, thus causing further counterfeits in the cluster private segment to be released after the BP has ensured the security of the derived clusters. After this, all transactions remain serially protected from composition and transitive composition attacks, and we further focus on the effects of correlation and critical absence in the global bag. Equation (Equation 2) shows the maximum counterfeits $h_{x}$ required for all overlaps $\Omega (C^{o})$ between subsequent releases in order to prevent from composition and transitive composition attacks [10]. Table 1 gives details on the symbols used in this paper.
(2)$$h_{x}\left(\Omega \left(C^{\theta }\right)\right)=\max_{O_{i}\in \Omega \left(C^{\theta }\right)}h_{x}\left(O_{i}\right)$$

K-anonymity and l-diversity fail to tackle dynamic data. Although they produce k-anonymous, l-diverse data, they only deal with one release at a time *i*, and due to the republication of data in the next release *j*, the chances of disclosure increase because they do not work on the relationship in i…j. These incremental updates need more protection. Due to the dynamic nature of data, we are also dealing with global bag privacy, and even after providing a distinct idea of the global bag, it is still vulnerable to assaults. Furthermore, there is still a link between the private and non-private segments after segregation. So, in Section 4.3, we will show how to improve the security of serial data.

### 4.3. Countering Correlation and Critical Absence Attack

In this study, we consider the data from the prisoners’ medical files where each prisoner was detained for various offenses. In our case, correlation attacks are possible due to the link between crimes and diseases. As shown in Table 5, if someone has prior information that $t_{2}^{*}$ a certain prisoner has been arrested for smoking in public areas, then given the presence of HIV and lung cancer in the said patient, the attacker can relate that the prisoner has lung cancer due to their smoking addiction. In this study, we look at various serial data attacks; one of our purposes is to guard against a correlation attack that involves many sensitive values. We present a novel framework for dealing with correlation attacks that will preserve the privacy in a better manner. Table 8 shows how we solved the correlation attack in [10], which was performed on transactional data by classifying non-sensitive properties in all releases. These offenses fall under many criminal classifications, with some of the most serious crimes, such as murder and manslaughter, which are classified as felonies. Some offenses, such as smoking, are personal, yet offenders were apprehended because they broke the law. Driving while under the influence of narcotics is also illegal. As a result, we believe that the answer that we have supplied is fascinating and helpful. This strategy does not need many resources, yet it is quite effective in preventing correlation attacks. The proposed technique can overcome correlation attacks between sensitive and non-sensitive variables by substituting a criminal category for a particular crime. Furthermore, if someone is arrested for transmitting AIDS to another person and is aware that they are afflicted, they may be prosecuted for unlawful HIV transmission. According to US law, this crime is a felony. We may improve the privacy of our data and make it simpler to safeguard them from data breaches by using this strategy for non-sensitive features.

Let us take a look at global bag assaults. While researching Sanony, we discovered the impact of the critical absence of sensitive values in the global bag, resulting in a serious privacy violation. Let us consider the example of herpes in Table 5 and Table 6, which indicate that an attack may be carried out due to the key lack of herpes in the global bag at timestamp 2. Absolute surjections between background knowledge and published data are required to rebuild data.

We presented the innovative idea of m-invariance [12] in the global bag to cope with critical absence attacks. M-invariance indicates that a signature stays the same in all releases to create m-diversity by producing counterfeits; however, it only functions for a single sensitive value. Our approach is unique in that we created the notion of counterfeits in the global bag to cope with numerous sensitive values. Sanony used backward and forward perturbation in the cluster private segment to reduce composition and transitive composition attacks by using the previously published corpus for BP and the freshly anonymized corpus for FP. Serial data are thus protected against composition and transitive composition attacks, although they are vulnerable to numerous global bag attacks. As previously stated, if the data on $t_{5}^{*}$ in Table 5 are removed, the attack may be readily carried out, as shown in Table 6. Another sort of attack on the global bag that we have seen is what happens if just one new record is added to a release with several sensitive values; how can one protect it if the adversary steals data from previously released data? We offered a technique by introducing the concept of counterfeits in the global bag in serial publishing and by examining the concept of privacy leaks based on an adversary’s prior information, as in critical absence attacks, as well as by introducing the concept of categorization in order to overcome correlations with other transactions that contain sensitive information as serial data, as in correlation attacks.

### 4.4. Privacy Analysis

Regarding the privacy analysis, we will go through the privacy risks associated with serial data in depth, taking past knowledge into account and skipping over privacy-preserving publishing that may be threatened by adding and deleting data. Dealing with dynamic data is challenging, yet it is commonly done in today’s context. We performed a comprehensive study of the theory of privacy and security in the global bag. Using examples from the privacy analysis, we will explain how an attacker can recreate microdata by integrating all publicly available data. With this information, one can determine whether or not a privacy violation is a possibility. The formulae and our method demonstrate why prior generalization rules failed and contributed to the development of the idea of the global bag’s so-called m−invariance. We demonstrate that the current model effectively lowers the risk of exposure. We provide a formal definition of privacy risk, which we will use to evaluate our privacy model by calculating the possibility of a transaction being compromised. We use surjective functions to construct theoretical ways of calculating risk due to background information.

**Definition** **11**(Privacy Risk)**.** *To reconstruct transaction t, an adversary will find reasonable surjections between background knowledge B(n) and $U^{*}\left(n\right)$. With Equation (Equation 3), the risk is estimated.*
(3)$$risk\left(t\right)=n_{breach}\left(t\right)/n_{total}$$
*$n_{total}$ is the number of reasonable surjective functions applied and $n_{breach}$ is the number of possible terms that reconstruct the private portion of t.*

If $n_{breach}\left(t\right)$ = $n_{total}$, then risk(t) is equivalently 1. If this condition is satisfied then an adversary with background knowledge can reconstruct sensitive values with a confidence of 100%. Surjective functions are reasonable and unreasonable because they can reconstruct original values or reconstruct fake values. This is why the risk is not the same for all values.

We will explain this with an example. Figure 3 shows the total reasonable surjections of a transaction $t_{5}^{*}$ at timestamp 1 and timestamp 2 because the adversary with the background knowledge knows that $t_{5}^{*}$ was deleted from timestamp 2. Figure 4 shows that two sensitive values are responsible for a data breach for $t_{5}^{*}$ and that two is the total number of reasonable surjective functions.

Equation (Equation 4) shows that if the surjective function is equal to the values of the data breaches, then the attacker B(n) finds the real sensitive value with 100% confidence.
(4)Risk(t)=2/2=1

When a record is deleted, such as that of $t_{5}^{*}$, and herpes is eliminated from the global bag in the next release, counterfeits are used in the global bag to protect the record of $t_{5}^{*}$, and the signature remains the same in all releases to protect them from a privacy breach.

Lemma 1 proves which tuple in $U^{*}\left(n\right)$ is the most vulnerable due to reconstruction in the next release.

**Lemma** **1.**
*For any transaction t and risk(t) = 1, there exists a single element in GSS(i) ∩ GSS(i+1) ∩⋯∩ GSS(j).*


Proofs of all lemmas are provided in Appendix A.

**Example** **2.**
*In this example, we explain the lemma by setting the lifetime of the transactions such that i⋯j = 2; Table 5 and Table 6 contain the records of two releases. The lifetime of $t_{5}^{*}$ is [1,1] because this global bag contains herpes and COVID-19 at time i, and at the time j, it contains diabetes and COVID-19. Therefore, Lemma 1 proves that the privacy breach risk of $t_{5}^{*}$ in two releases is one—an adversary can easily reconstruct the sensitive data.*


**Lemma** **2.**
*$\cap_{k=i}^{n}=t.GSS\left(k\right)$ contains all sensitive values of t, which are reconstructed from reasonable surjective functions.*


In terms of providing protection, Lemma 2 proves that, when an adversary obtains information about $t_{1}^{*},t_{2}^{*}\cdots t^{*}\left(n\right)$ from $U^{*}\left(n\right)$ and has previous knowledge B(n), they may conclude that the real sensitive value surely falls within GSS(i + 1) ∩⋯∩ GSS(j). This set is large enough for protection in the presence of republication until a condition is reached in which transaction *t* is deleted, and it cannot be compromised after deletion. To achieve this strategy and to protect the global bag, we need to explain the concepts of m-invariance and m-uniqueness in the global bag. All values in the global bag must appear at most one time. Ignoring this results in the sensitive values of many transactions. This is the concept of m-uniqueness. However, in the case of dynamic data and republication, all releases containing the same set of sensitive values are the cure for critical absence through the maintenance of m-invariance in a global bag of all releases. Let there be a transaction *t* ∈ $U^{*}\left(n\right)$ with lifetime (i,i+1,⋯,j−1,j). We say that it is m-invariant if it contains the same set of sensitive values throughout the lifespan (i⋯j). According to Example 2, the global bag of the two releases shown in Table 5 and Table 6 is m-unique, but not m-invariant. We need to provide a solution to this problem by introducing counterfeits because, without counterfeits, the m-invariant condition is violated, which means that the signature is the same in all releases.

**Definition** **12**(Counterfeit Requirement)**.** *Counterfeits (Tc) originate from terms that have already been used within a release. The terms are taken from private terms that have already been used in clusters to reduce utility loss when introducing counterfeits. They are required when we need to protect every term from breaches, but the counterfeits must not match with any terms in the global bag.*

**Lemma** **3.**
*If a serial publication of $E^{*}=T_{1}^{*},T_{2}^{*}\ldots T^{*}\left(n\right)$ follows Kanony, then the risk of t at any given time i must be less than 1/k.*

(5)
$$Risk(t_{i})<=1/k$$

*The signature must be the same in a global bag of all releases to achieve this. The signature remains constant in all releases to prevent attacks due to the critical absence of herpes. We are dealing with serial data in which each previous year’s record is published, and the adversary can take information from these data. So, there is a need to maintain the signatures in all releases to prevent attacks. For any t ∈ $U^{*}\left(n\right)$, the risk(t) is given in the proof of Equation (Equation 5) in Lemma 3.*


A discussion attack occurs when a particular record is added to the global bag and its signature differs from that in previously published releases.

**Example** **3.**
*There is a problem of addition in the records in Table 8 and Table 9, such as in the $t_{7}^{*}$ record. Due to this, a single value (diabetes) added into the global bag can allow the adversary to breach the data of $t_{7}^{*}$. As the previous record has been published, to provide privacy to $t_{7}^{*}$, we have to add a counterfeit into that release. By inserting a counterfeit into the global bag of that release, we may protect this record.*


Here, the concept of unreasonable surjection arises. We used surjective functions to maximize the adversary’s information in order to calculate this likelihood. However, sometimes, adversaries can calculate unreasonable surjections, which can be explained by looking at Table 8 and Table 9. Suppose that, as shown in Table 10, the counterfeit is removed due to the addition of more records; the attacker may thus map a disease to the record of a deleted individual. In that case, it is an unreasonable surjection. Hence, this proves that the risk factor is not the same for all records. Eventually, we resolve this issue, which is our aim.

Lemma 3 discusses the suggested publishing mechanism for dealing with the issue of privacy-preserving serial publication.

With the anonymizing technique [11], we partitioned data into the private, non-private, and global bags. We gave a solution to the problems that arose in [10]; if a correlation attack can be performed due to correlation, then categorizing the non-private portion is the solution according to Algorithm 1. Attacks arising from the critical absence of sensitive values can be overcome by the novel concept of introducing counterfeits into the global bag. We introduced herpes as a counterfeit in all releases so that the data of $t_{5}^{*}$ could not be breached. The probability of herpes is 1/x (x = overall number of transactions in a corpus), thus preventing attacks due to critical absence. When we continuously add data along with the deletions, we see that if a new record is added into the global bag, it is not easy to protect these data because the previous record has already been published. Therefore, we protect them by adding a group of counterfeits into the global bag to decrease the risk of attack according to Algorithm 2. Ensuring m−invariance in all releases is crucial for privacy-preserving during the republication of the data.
**Algorithm 1:** Kanony(a) **Input:** Crimes *c* in corpora $E^{\wedge }$ **Output:** Categorized $c^{*}$ **1.** Initialize = [list1],[list2],[list3]; **2.** for [i],[j],[k] in crimes **3.** If crime[i]=x, then **4.**      [i]⟵[list1] **5.** elseif **6.**      [j]⟵[list2] **7.** else **8.**      [k]⟵[list3] **9.** end for **10.** Return $c^{*}$ in corpora $E^{\wedge }$

**Algorithm 2:** Kanony(b) **Input:** A preserving corpus $T^{\wedge }$, the anonymized corpus $T_{n+1}^{*}$ **Output:** The serially preserving corpora $E_{n+1}^{\wedge }$ **1.** For every anonymized cluster $C^{*}\in T_{n+1}^{*}$ **2.** If critical absence = 0, then (Definition 9) **3.**      return $C^{*}\in T^{*}$ **4.** Else **5.**      $C^{*}$⟵ Call counterfeits $T_{c}$ **6.** for all transactions $t^{*}\in T_{n+1}^{*}$ do **7.**      $t^{*}$⟵ Call counterfeits $T_{c}$ (Definition 11) **8.** end for **9.** Return $T_{n+1}^{\wedge }$

## 5. Experimental Evaluation

This section describes experiments that were performed to check records that were vulnerable due to the critical absence of sensitive values in the global bag, perturbation rates, and the effects on utility in terms of query count and performance.

### 5.1. Experimental Setup and Datasets

We used the real-world BMS Webview-1 (B1) and BMS Webview-2 (B2) datasets for our experimental assessment. Table 11 show details of datasets used in our experimentation. To evaluate the implementation, it was run on an Intel Core i5 2.50 GHz laptop with 8 GB of RAM and the Windows 10 operating system, with the PyCharm platform being used for the assessment.

Clickstream data from online stores were included in this dataset. Two algorithms were developed and implemented. (1) Sanony addresses assaults on serial data, but the global bag remains susceptible to flaws. (2) Kanony is our technique that addresses all of the attacks that are also addressed by Sanony, and it further protects the global bag against critical absence attacks and correlation attacks that use sensitive and non-sensitive attributes. It is necessary to employ a data generator that has configurable parameters, such as sample size (sz), to build serial corpora sequentially from a vast corpus of transactional information. A total of five years of serial corpora—$T_{1},T_{2},T_{3},T_{4},$ and $T_{5}$—were employed in the evaluation. We worked on datasets with 50,000 and 70,378 transactions. To conduct the experiments, we compared Kanony and Sanony using two datasets to see which one performed better. We carried out the comparison by adjusting various parameters, such as the following:Sample size: The number of transactions *t* that we see in a serial corpus *T* with a sample size ranging between 1.25% and 10% and releases ranging from 1 to 5.Releases: These are, in fact, the timestamps at which the data are released to the public. We must be very careful because we are dealing with serial data, which are continuous, use both public and private terms, or contain data that are regularly added and removed. Each of the various sample sizes ranged between 1.25% and 10%, with timestamps ranging from 1 to 5. Our experiments were carried out on both of the aforementioned datasets. We compared our technique with Sanony, which is similarly concerned with serial data.

### 5.2. Critical Absence Effect

These experiments explain the possibility of an attack occurring due to the essential lack of sensitive values in the global bag. The number of transactions at risk due to the critical absence effect was estimated for each anonymized corpus in $T_{n}$, and the number of susceptible records was graphically represented. However, due to various circumstances, the risks associated with each transaction were not the same. For this reason, we created graphs using a variety of sample sizes and releases, each with a unique set of outcomes. More transactions are at risk as the amount of susceptible data grows, perhaps resulting in a serious privacy breach. An illustration of the number of transactions that are at risk with respect to varying sample sizes due to the critical absence of sensitive values in the global bag is shown in Figure 5.

Each pair of lines represents the two different techniques, and the graph shows the number of transactions that are at risk with respect to the varying sample sizes. For all experiments, the BMS Webview-1 B(1) and BMS Webview-2 B(2) datasets and the Kanony and Sanony techniques were used. According to the differences measured in the results, the number of transactions in the global bag of Sanony for serial publishing was at risk due to the critical lack of sensitive values in the global bag; however, there were no susceptible records detected in the global bag when using our technique. We ran tests with varying sample sizes of 1.25%, 2.5%, 5%, and 10% and plotted graphs for the vulnerable records for both datasets. When we increased the sample size, we observed that the number of susceptible records in the global bag grew equally for both datasets, indicating that Sanony became more vulnerable. As a result, many transactions were still at risk, even after the solution offered by Sanony was implemented. Sanony did not work for protecting the privacy of the global bag; instead, it solely focused on the security of cluster private segments. Our approach for these sensitive records is to use counterfeits in the global bag to protect them. We saw increased protection as a result of the Kanony strategy. Hence, the experimentation proved that adding counterfeits into global bags reduced the risk of data disclosure, thus preventing attribute disclosure attacks on serial data.

As shown in Figure 6, we performed experiments and calculated the vulnerable records at different timestamps ranging from 1 to 5 for both datasets (BMS Webview-1 B(1) BMS Webview-2 B(2)). In these consecutive timestamps of the serial data, records were deleted. As a result, there were greater chances of the critical absence of sensitive values at each subsequent timestamp than in the previous release for both datasets. Moreover, we also observed that B(2) had more vulnerable records than B(1) because of the large size of the B(2) dataset. Hence, the experimentation proved that vulnerability increased upon subsequent releases in the case of Sanony because more records were deleted in the next release compared to the previous ones, and Kanony catered to this problem by using counterfeits in the global bag.

### 5.3. Perturbation Rate

We conducted experiments, computed the perturbation rates (percentage of counterfeits) for our approach, and compared them to those of Sanony [10] to determine which is more suitable in terms of privacy. The number of counterfeits varied depending on various criteria, such as sample size and timestamps. We took sample sizes of 1.25%, 2.5%, 5%, and 10% in order to calculate the perturbation rates for the global bag in the case of Kanony, and the same breaking points were used to calculate the overall perturbation required by Kanony and Sanony in the private segment. The perturbation rate was defined as the number of counterfeits necessary to achieve the appropriate level of privacy protection. The experiments showed that our technique beat the competition on both datasets, with a small increase in the perturbation rate. This is because Kanony needs counterfeits to protect the global bag, which was vulnerable to assault in earlier methods. We would like to make the global bag more confidential, and we focus on computing the global bag’s perturbation rates in all releases, which are presently disregarded by Sanony.

As shown in Figure 7, in proportion to the increase in sample size, there was an increase in the likelihood of a critical absence of sensitive values, resulting in the need for additional counterfeits to record these transactions. We showed that B(2) had a higher perturbation rate than that of B(1) when they were compared. In all datasets, the perturbation rate of the global bag fluctuated depending on the sample size used in the experiment. In actuality, both datasets are real-world datasets, and since data vary in nature, there was some variation in the results, as we saw with the sample size of 2.5%, whose value in B(1) was greater than that in B(2). Moreover, we did not compare the perturbation rate of Kanony with that of Sanony, since prior techniques did not work on the privacy of the global bag, but our methodology works on the privacy of the global bag. According to the graphs in the section on the critical absence effect, in the case of Sanony, the vulnerability increased upon subsequent releases because more records were added and deleted in the next release compared to the previous release, and Kanony caters to this problem by using counterfeits in the global bag.

Figure 8 shows the perturbation rates of the global bag required by B(1) and B(2) in all releases, from 1 to 5. As we can see in the graphs, too little perturbation was needed, and as a result, security was provided to all of the serial corpora ($T_{1},T_{2},T_{3},\cdots ,T_{n}$). Because of the increase in sample size, B(2) had a slightly higher perturbation rate than that of B1, which increased the likelihood of being vulnerable to an attack. Furthermore, to secure the privacy of a single transaction, we must include a counterfeit in the release to protect its record, since earlier releases have been made available for viewing. If a key absence happens, the adversary may access vital information. Therefore, adding a small number of counterfeits into a global bag provides a stronger privacy guarantee for serial data than Sanony does.

Figure 9 presents a comparison graph that shows the total amount of perturbation that is necessary, as represented by two lines. Kanony is represented by a blue line, while a red dotted line represents Sanony. The two lines represent Sanony and Kanony for datasets B(1) and B(2). The perturbation rates of both the global private and cluster private segments for Kanony and Sanony are shown in Figure 9. Even though Sanony protected the cluster private segment against composition attacks, no progress was made on the security of the global bag. With the addition of just a tiny number of counterfeits into the global bag, Kanony could respond to assaults that arose in the global bag. The graphs show only the marginal difference in perturbation rate for both approaches, and the slight increase for Kanony is due to the counterfeits required in the global bag. This shows that, due to the difference in sample size, B(1) had a lower perturbation rate of 0.21%, on average, compared to B(2), which had 0.512%.

### 5.4. Utility

This section analyzes the minimal impact of adding counterfeits in Kanony for actual queries by comparing it with Sanony. Kanony introduces the concept of counterfeits in the global bag, and due to the categorization of the non-private set, it affects utility. We will not directly compare our method to Sanony, but rather demonstrate a trade-off in utility loss when utilizing counterfeits in the global bag and categorizing non-sensitive attributes to meet the privacy guarantee. Following the perturbation rates, we will determine the error levels of Sanony and Kanony. We first anonymized the serial data $T_{1},\cdots ,T_{5}$ and made groups in datasets based on term associations. Next, the original and anonymized groups were selected as queries from the dataset. Then, we calculated the error rate. The relative error was used to calculate the utility loss of an association (a,b).
(6)re=abs(so(a,b)−sp(a,b))/AVG(so(a,b),sp(a,b))
where abs() yields the absolute value and so(a,b) and sp(a,b) are the supports of (a,b) in the original and anonymized datasets, respectively. False associations that did not exist in the original corpus were captured. To prevent division by 0, the denominator utilized an average rather than the original value when bogus transactions occurred. Different queries were used to compare the data, and the results were averaged to achieve greater accuracy.

We calculated the utility at different sample sizes of 1.25%, 2.5%, 5%, and 10%, as shown in Figure 10, which contains reconstructed transactions because the entries are randomly linked in the vertical segment of the cluster. For each query, we compared anonymized data in which data were generated randomly by linking entries with original data. As the data were dynamic and due to the addition and deletion of records, the cluster formation changed at each timestamp, and the values were altered in the private segment. The results showed only marginal differences. The purpose of this section is to see the trade-off in how much Kanony degrades usefulness compared to Sanony due to the counterfeits in the global bag and the generalization for dealing with correlation attacks. However, at the same time, Kanony provides a greater privacy guarantee than Sanony.

### 5.5. Time Consumption

The experiments in this section show the performance of Kanony. In this section, we will compare the efficiency of Kanony and Sanony by calculating their time consumption. For both datasets (BMS Webview-1 (B(1)) and BMS Webview-2 (B(2))), we took sample sizes of 1.25%, 5%, and 8.75%. Figure 11 shows two plots and the observed performance of Kanony and Sanony for datasets B1 and B2 containing transactions $t_{1}^{*},t_{2}^{*},t_{3}^{*}\cdots t^{*}\left(n\right)$.

By looking at Figure 11, we observe that because there were more dynamically inserted tuples when the sample size was increased, the runtime rose with respect to the sample size. Because both approaches used the same anonymization technique, there were no substantial differences in runtime. As a result of the fact that we utilized real-world datasets, we saw a little increase in Sanony’s runtime. This was because they exhibited somewhat different behaviors at certain moments. When we see the privacy guarantee provided by Kanony, we think that the computational cost is a reasonable price to be paid. However, work may be done in the future to reduce the computing cost of the algorithm. Furthermore, because of the big size of B(2), it had a greater computational cost due to the vast number of transactions it needed to process. High computing costs are a possible disadvantage of most privacy-providing systems, which is true in our situation.

## 6. Conclusions

Microdata are not protected during republication using current centralized publication mechanisms. Continuous release of transactional data for data analytics might lead to serious privacy violations. Serial publishing scenarios with recurring data releases, such as those with prisoner health information, exacerbate the problem. We provide an alternative comparative privacy measure to meet the privacy claims with this approach. Our solution overcame the challenges of extending stringent privacy assurances for serial data in a transactional form. We investigated the privacy-preserving serial publishing of transactional data in this paper. The privacy analysis section employed surjective functions to calculate the attack probability in order to offer some theoretical justification. Surjective functions were used to determine the likelihood of an attack based on an adversary’s prior information. We looked at the possibility of correlation between sensitive and non-sensitive variables in more detail. We saw that a correlation attack can only be carried out if sensitive and non-sensitive data are linked. This paper also tackled the critical absence of values when dealing with multiple sensitive values by introducing a unique idea that precludes an adversary from inferring sensitive data from a global bag’s numerous releases. Our scheme was to use counterfeits to reduce the chances of sensitive terms being left out of the global bag. We used this deception strategy to protect privacy in serial publication. This paper offered a comprehensive experimental investigation to demonstrate its theoretical framework, showing that current publications may be serially preserved by using this approach. This indicates that privacy in the global bag is retained even after the addition and critical absence of data. We presented Kanony to protect microdata from various threats. A serial publishing scenario in which several sensitive values from the same person are disclosed at the same time, such as the yearly release of prisoner health data [1], creates a difficulty. We first suggested the concept of m−invariance to create a guarantee of privacy in the global bag. Then, by expanding this guarantee, we could lower the risk of data breaches (correlation attacks and critical absence). In our case, the adversary’s posterior information and knowledge about the non-sensitive portion of the transaction are required for breaches [10]. A crucial lack of sensitive values or a sudden record addition can result in attacks on various global bag releases. Changes in signatures enhance the likelihood of an attack. To prevent such assaults, we created a counterfeiting strategy. Counterfeits may protect people’s records that contain sensitive terms. We showed that when fewer counterfeits are introduced, the utility does not deteriorate much. Finally, we used real datasets to evaluate the efficacy of our technique compared to that of state-of-the-art methods. Serial publishing systems, in particular, are vulnerable to correlation attacks, which our approach effectively defends against. Compared to earlier approaches, our model was shown to be more successful in detecting privacy breaches and enhancing privacy. The drawback that we found with this strategy was that it causes a tiny increase in relative error, but it also improves privacy compared to earlier approaches. Using a prisoner’s health record as an example, we paid close attention to and spotted critical attacks in serial data. Serial data are dynamic, which is why these attacks are possible, and we live in an era in which dynamic data are prevalent. Our technique is quite broad in scope and may be used on various forms of serial data. There are various sorts of data for which we can utilize the techniques described in this article, such as bank records in which there are multiple sensitive values of an individual, such as their salary when performing two jobs at the same time, or a bank account holder who has two accounts at the same time, as well as personal profiles and business data. This work also paves the way for several thrilling future directions in resolving the issues of serial data.

### Future Work

In our proposed scheme, we work on a global bag and protect a person’s sensitive data from attack due to the critical absence of sensitive values. What happens if a sensitive record is not deleted? It will be updated in the global bag instead—for example, when the same patient suffers from two different diseases at different times. We will work to protect from this type of attack on privacy in the future. Moreover, we will improve the utility of data by maintaining privacy.

## Figures and Tables

**Figure 1 sensors-22-02811-f001:**
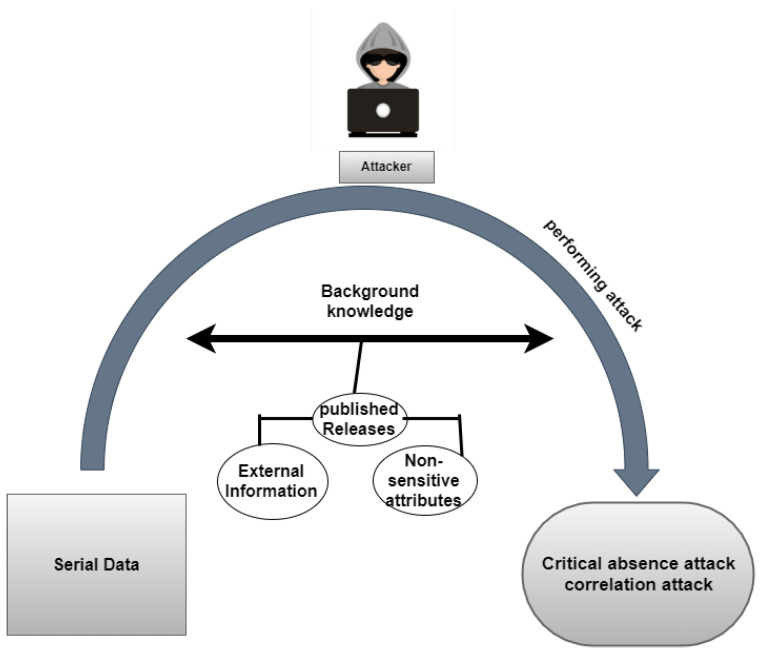
Attacker model.

**Figure 2 sensors-22-02811-f002:**
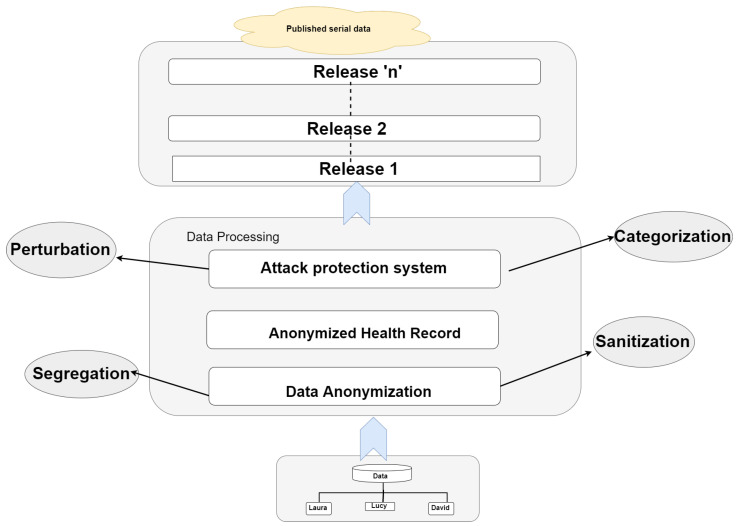
Proposed model.

**Figure 3 sensors-22-02811-f003:**
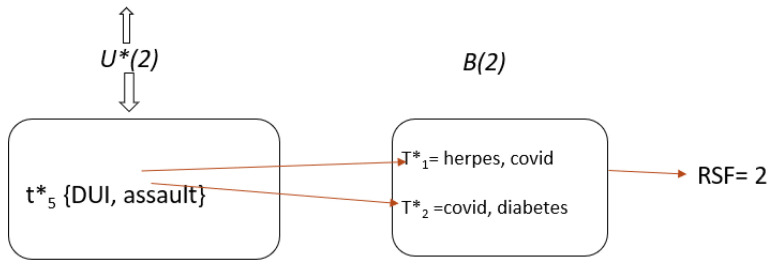
Reasonable surjections.

**Figure 4 sensors-22-02811-f004:**
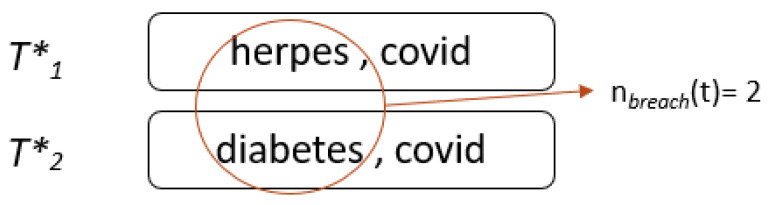
Sensitive values responsible for a breach.

**Figure 5 sensors-22-02811-f005:**
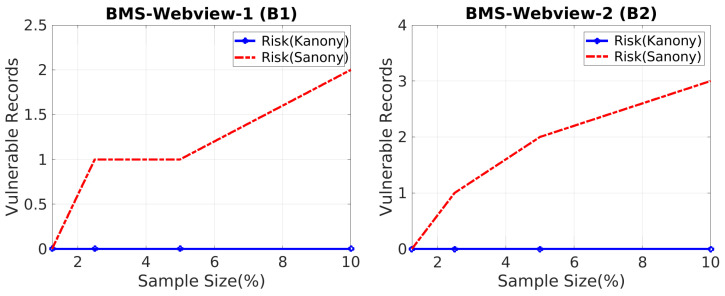
Vulnerability vs. sample size.

**Figure 6 sensors-22-02811-f006:**
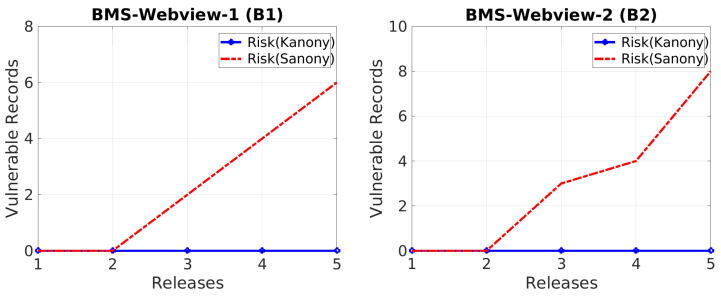
Vulnerability vs. releases.

**Figure 7 sensors-22-02811-f007:**
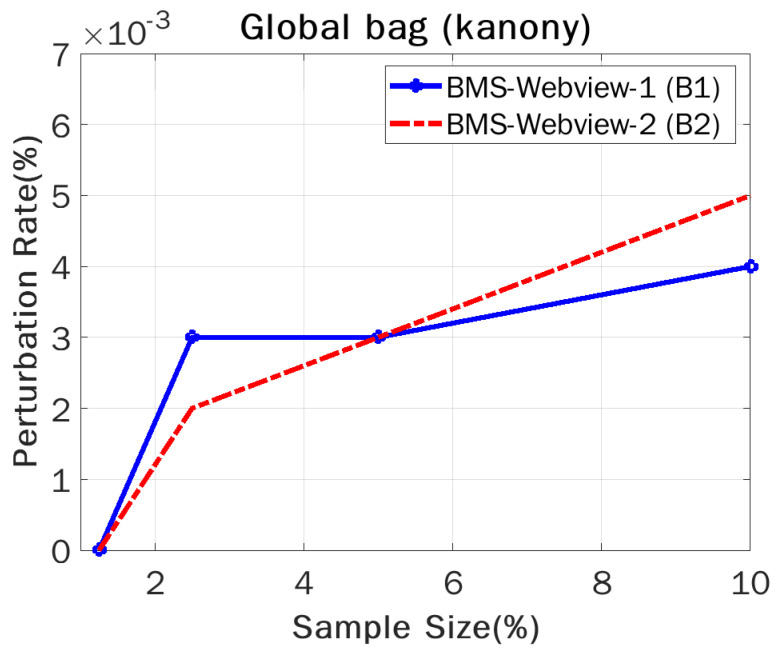
Perturbation rate vs. sample size.

**Figure 8 sensors-22-02811-f008:**
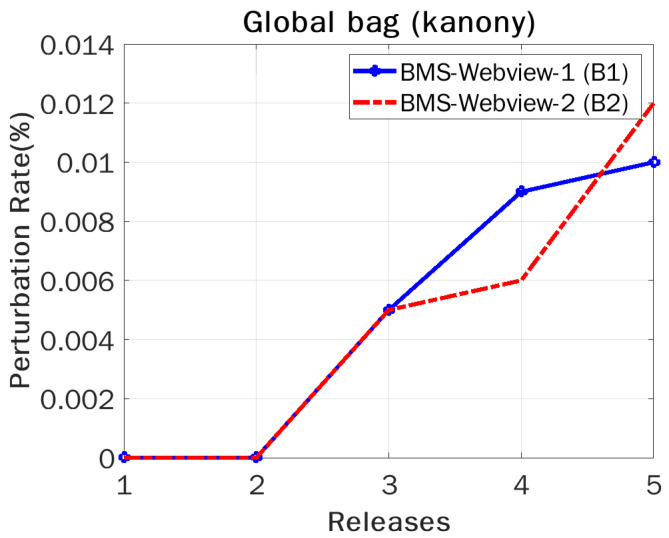
Perturbation rate vs. release.

**Figure 9 sensors-22-02811-f009:**
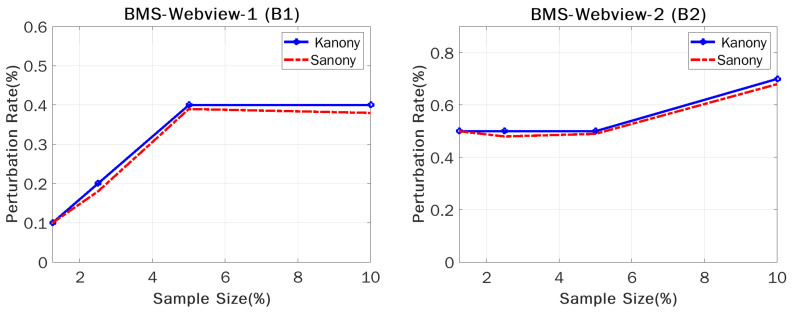
Perturbation rate vs. sample size.

**Figure 10 sensors-22-02811-f010:**
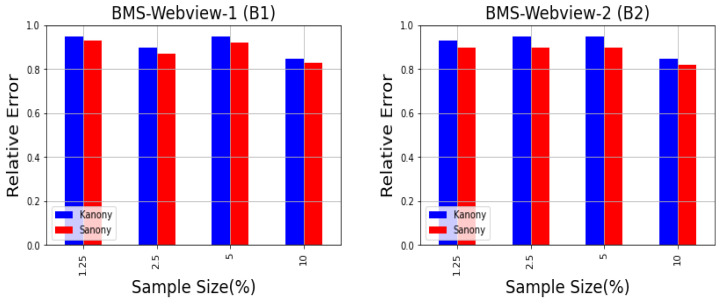
Utility vs. sample size.

**Figure 11 sensors-22-02811-f011:**
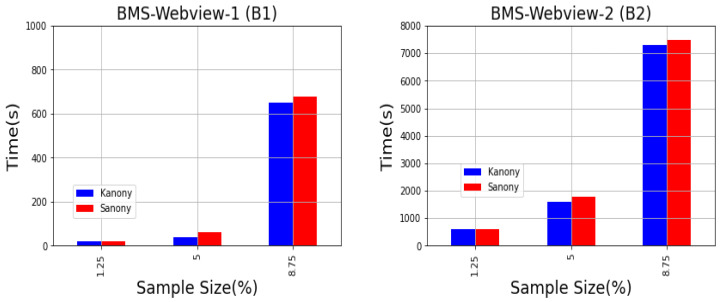
Runtime vs. sample size.

**Table 1 sensors-22-02811-t001:** Notations.

*t*	set of terms known as a transaction
*C*	subset of transactions known as a cluster
*T*	set of all transactions in one timestamp known as a corpus
*E*	known as serial corpora
$T_{s}$	all private term sets in corpora
${\bar{T}}_{s}$	all non-private term sets in corpora
$T_{g}$	set of all private terms in the global bag
$U^{*}\left(n\right)$	union of anonymized transactions at time *n*
B(n)	background knowledge at time *n*
Tc	counterfeits required for transactions
$O_{i}$	any overlap belongs to $\Omega (C^{o})$
$\Omega (C^{o})$	set of overlapping transactions
$h_{x}$	maximum counterfeits for all overlaps
[t,T,C,E]	original version
$\left[t^{*},T^{*},C^{*},E^{*}\right]$	anonymized version
$\left[t^{\wedge },T^{\wedge },C^{\wedge },E^{\wedge }\right]$	final released version

**Table 2 sensors-22-02811-t002:** Serially generated prisoner health records (Release 1).

Name	Record
Laura	$t_{1}$ theft, arson, fraud, HIV
Lucy	$t_{2}$ theft, arson, smoking, lung cancer
Martin	$t_{3}$ vandal, arson
Shane	$t_{4}$ abuse, arson, cancer, COVID-19
John	$t_{5}$ DUI, assault, HIV, herpes
Stacy	$t_{6}$ DUI, assault

**Table 3 sensors-22-02811-t003:** Serially generated prisoner health records (Release 2).

Name	Record
Laura	$t_{1}$ theft, arson, fraud, HIV
Lucy	$t_{2}$ theft, arson, smoking, lung cancer
Martin	$t_{3}$ vandal, arson
Shane	$t_{4}$ abuse, arson, cancer, COVID-19
John	$t_{5}$ DUI, assault, HIV, herpes
Stacy	$t_{6}$ DUI, assault
Ben	$t_{7}$ theft, arson, murder, lung cancer, diabetes
Ivy	$t_{8}$ abuse, arson, mansl.

**Table 4 sensors-22-02811-t004:** Serially generated prisoner health records (Release 3).

Name	Record
Laura	$t_{1}$ theft, arson, fraud, HIV
Lucy	$t_{2}$ theft, arson, smoking, lung cancer
Martin	$t_{3}$ vandal, arson
Shane	$t_{4}$ abuse, arson, cancer, COVID-19
John	$t_{5}$ DUI, assault, HIV, herpes
Stacy	$t_{6}$ DUI, assault
Ben	$t_{7}$ theft, arson, murder, lung cancer, diabetes
Ivy	$t_{8}$ abuse, arson, mansl.
Pam	$t_{9}$ abuse, arson, mansl., cancer
Jan	$t_{10}$ theft, arson, HIV

**Table 5 sensors-22-02811-t005:** Anonymized prisoner health records (Release 1).

${\bar{T}}_{s}$	$T_{s}$	$T_{g}$
Cluster 1		herpes, COVID-19
$t_{1}^{*}${theft, arson, fraud}	HIV	
$t_{2}^{*}${theft, arson, smoking}	lung cancer	
Cluster 2		
$t_{3}^{*}${vandal, arson}	cancer	
$t_{4}^{*}${abuse, arson}		
Cluster 3		
$t_{5}^{*}${DUI, assault}	HIV	
$t_{6}^{*}${DUI, assault}		

**Table 6 sensors-22-02811-t006:** Anonymized prisoner health records (Release 2).

${\bar{T}}_{s}$	$T_{s}$	$T_{g}$
Cluster 1		COVID-19, diabetes
$t_{2}^{*}${theft, arson, smoking}	lung cancer	
$t_{7}^{*}${theft, arson, murder}	HIV	
Tc		
Cluster 2		
$t_{3}^{*}${vandal, arson}		
$t_{4}^{*}${abuse, arson}	cancer	
$t_{8}^{*}${abuse, arson, mansl.}		
Tc		

**Table 7 sensors-22-02811-t007:** Anonymized prisoner health records (Release 3).

${\bar{T}}_{s}$	$T_{s}$	$T_{g}$
Cluster 1		diabetes
$t_{2}^{*}${theft, arson, smoking}	lung cancer	
$t_{7}^{*}${theft, arson, murder}	HIV	
$t_{10}^{*}${theft, arson}		
Cluster 2		
$t_{8}^{*}${abuse, arson, mansl.}		
$t_{9}^{*}${abuse, arson, mansl.}	cancer	

**Table 8 sensors-22-02811-t008:** Published prisoner health record (Release 1).

${\bar{T}}_{s}$	$T_{s}$	$T_{g}$
Cluster 1		herpes, COVID-19
$t_{1}^{*}${property, personal crime, felonies}	HIV	
$t_{2}^{*}${property, personal crime, felonies}	lung cancer	
Cluster 2		
$t_{3}^{*}${personal crime, felonies}	cancer	
$t_{4}^{*}${misdemeanors, felonies}		
Cluster 3		
$t_{5}^{*}${criminal offense, felonies}	HIV	
$t_{6}^{*}${criminal offense, felonies}		

**Table 9 sensors-22-02811-t009:** Published prisoner health record (Release 2).

${\bar{T}}_{s}$	$T_{s}$	$T_{g}$
Cluster 1		{COVID-19, diabetes (Tc herpes, AIDS)}
$t_{2}^{*}${property, personal crime, felonies}	lung cancer	
$t_{7}^{*}${property crime, felonies}	HIV	
Tc		
Cluster 2		
$t_{3}^{*}${personal crime, felonies}		
$t_{4}^{*}${misdemeanors, felonies}	cancer	
$t_{8}^{*}${misdemeanors, felonies}		
Tc		

**Table 10 sensors-22-02811-t010:** Published prisoner health record (Release 3).

${\bar{T}}_{s}$	$T_{s}$	$T_{g}$
Cluster 1		{diabetes (Tc herpes, COVID-19, AIDS)}
$t_{2}^{*}${property, personal crime, felonies}	lung cancer	
$t_{7}^{*}${property crime, felonies}	HIV	
$t_{10}^{*}${property crime, felonies}		
Cluster 2		
$t_{8}^{*}${misdemeanors, felonies}		
$t_{9}^{*}${misdemeanors, felonies}	cancer	

**Table 11 sensors-22-02811-t011:** Description of the datasets.

Datasets	No. of Trans.	No. of Terms	Max Trans. Length	Avg. Trans. Length	Avg. Sparsity
B1	50,000	497	267	2.5	99.49%
B2	70,378	3340	161	5.0	99.86%

## Data Availability

Not applicable.

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
