# Peer review of "A Novel Privacy Paradigm for Improving Serial Data Privacy"

_sensors, 2022, doi:10.3390/s22072811_

Round 1

Reviewer 1 Report

This paper proposes a privacy-preserving approach for serial data publishing as well as data re-publishing. By anonymizing and generalizing the data, and adding perturbations to the "global bag", the proposed approach may defend against correlation and critical absence attacks, and hence improve the security of the global bag.

However, the contribution of this work is not substantial. The authors' approach tends to be more complementary to the shortcomings of the existing work (especially reference 9). Among the four contributions mentioned in the article (page 5, line 3), "Protecting global bag in the serial publication and Minimizing risk factor" is one of the more attractive points and should be highlighted. This part and its research significance should be highlighted. The remaining two contributions, “Dealing with correlation attack and Preserve individual privacy”, are not innovative in their solutions. In addition, the length of the article is very long, but it does not clearly express its core work and the difference with others' work, so it is suggested to streamline the content of the article.

A few suggestions and comments for improvement:

  1. In the introduction (page 2, line 16), the authors should detail the problems that exist in the reference [9], for example, the article mentions (page 2, line 17) "but still some attacks exist", more details should be specified on the mentioned attacks.

  1. The motivation subsection is suggested to be divided into several paragraphs.

  1. Some charts in the article are redundant. For example, Tables 1-3 are only meant to express continuous data generation and could be represented in a single table entirely (either with a dashed line separating the updated data or some other more concise way). The rest of the tables in the article are also.

  1. In the section Definition 3 (Background Knowledge) of Preliminaries, what is the meaning of “At the time n≤1”?

  1. In Section 4, Proposed Methodology, the authors claim to be more concerned with the security of the "global bag", which is not described in detail in Subsections 4.1 and 4.2. In subsection 4.3, the authors only state how to resist two types of attacks (i.e., replacing crime categories to solve correlation attacks and adding counterfeits to resist critical missing attacks.) The authors should present their approach in detail and formally, and analyze the impact of the approach on the security of the "global bag" in a comprehensive manner. This section does not present the innovative nature of their approach very well.

  1. It is suggested to present the privacy analysis of the method in a separate subsection. Also, the conclusion section is too verbose, just focusing on expressing the contribution of the work and the results obtained.

  1. The organization and language need to be revised for readability. The article does not express the author's work concisely. There is too much superfluous content. In addition, attention should be paid to the grammar and formatting of the article. (For example, page 6, line 11, "is know" should be revised to "is known", page 6, line 13 "At time n ≤1", missing space. "Page 7, Definition 8, "Let t* belongs" should be revised to "Let t* belong", etc.)

Author Response

A Novel Privacy Paradigm for Improving Serial Data Privacy

Ayesha Shaukat, Adeel Anjum, Saif U. R. Malik, Munam Ali Shah, Carsten Maple

We would like to thank the reviewer for the valuable feedback and suggestions. We believe that incorporating these comments have significantly improved the quality of the work. Following is the point-by-point response to the comments.

Reviewer 1

Comments and Suggestions for Authors

This paper proposes a privacy-preserving approach for serial data publishing as well as data re-publishing. By anonymizing and generalizing the data, and adding perturbations to the "global bag", the proposed approach may defend against correlation and critical absence attacks, and hence improve the security of the global bag.

However, the contribution of this work is not substantial. The authors' approach tends to be more complementary to the shortcomings of the existing work (especially reference 9). Among the four contributions mentioned in the article (page 5, line 3), "Protecting global bag in the serial publication and Minimizing risk factor" is one of the more attractive points and should be highlighted. This part and its research significance should be highlighted. The remaining two contributions, “Dealing with correlation attack and Preserve individual privacy”, are not innovative in their solutions. In addition, the length of the article is very long, but it does not clearly express its core work and the difference with others' work, so it is suggested to streamline the content of the article.

Response

  • Thank you for your insightful remarks. Thank you for taking the time to look at our manuscript. We've modified our document in response to reviewer comments and recommendations.
  • The proposed work's major goal is to protect the privacy of serail data by protecting from various attacks and global bag protection is one of them. According to reviewer recommendations we have improved our work. Actually all the four contributions are of great importance. We have explained them further for clear understanding. Dealing with correlation attack is not innovative but we are working with multiple sensitive values in serial data and preventing from correlation attack while maintaining privacy of multiple sensitive values in serial data is complex task. In order for clear understanding we have highlighted all the contributions in manuscript in contribution section.
  • As our one of the contribution says protecting individual privacy, it means that if only single record is deleted then we can still provide privacy to that individual record by using our technique. This is actually a plus point of our approach. According to the comments, the introductory section, methodological,literature and experimental all sections have been revised by adding more relevant details.

A few suggestions and comments for improvement:

  • In the introduction (page 2, line 16), the authors should detail the problems that exist in the reference [9], for example, the article mentions (page 2, line 17) "but still some attacks exist", more details should be specified on the mentioned attacks.
  • Thank u so much for guiding us. In our manuscript we have highlighted our point of “some attacks” in detail. However, there are still several assaults on which we shall focus in our study for example critical absence in global bag. As we all know, dealing with serial data is a difficult process due to its dynamic nature. Attacks on sanony have been discovered due to the crucial lack of sensitive values and the link between sensitive and non-sensitive attributes. It's critical to raise awareness of these assaults and discover a suitable way to combat them to make serial data more secure.
  • The motivation subsection is suggested to be divided into several paragraphs.
  • We are delighted to thoroughly examine the text. In the updated document, we have made all of the necessary adjustments and enhancements. Motivation section of the new document has been enhanced in terms of structure, paragraphs writing flow.
  • Some charts in the article are redundant. For example, Tables 1-3 are only meant to express continuous data generation and could be represented in a single table entirely (either with a dashed line separating the updated data or some other more concise way). The rest of the tables in the article are also.
  • We are thankful on your review. We have distinguished all the tables in manuscript to increase readability by assigning this continuous data generation as release1, release2 guided by Reviewer 2 and same style is used in past papers we have studied. I assure you that now it is easily understandable.

  • In the section Definition 3 (Background Knowledge) of Preliminaries, what is the meaning of “At the time n≤1”?
  • We sincerely apologise for the inconvenience, which was caused by a typo mistake that will not be visible in the revised version. It is truly at time 'n,' which indicates that at any point in time, the adversary's information about the suspect, which we designated by the letter 'n,' is available.
  • In Section 4, Proposed Methodology, the authors claim to be more concerned with the security of the "global bag", which is not described in detail in Subsections 4.1 and 4.2. In subsection 4.3, the authors only state how to resist two types of attacks (i.e., replacing crime categories to solve correlation attacks and adding counterfeits to resist critical missing attacks.) The authors should present their approach in detail and formally, and analyze the impact of the approach on the security of the "global bag" in a comprehensive manner. This section does not present the innovative nature of their approach very well.

  • We will take into consideration all of the insightful remarks you provided when examining our text. The paper will be improved as a result of your useful ideas, which we are grateful to receive. An update to the relevant section has been made to reflect the above-mentioned improvement. In reality, we are giving protection against a wide range of assaults that might emerge while dealing with serial data. These assaults are discussed in detail in Sections 4.1 and 4.2. Furthermore, in order to fight assaults on the global bag, we must first establish the global bag itself. We have highlighted the concern of global bag and anonymization technique dicuss in section 4.1 in manuscript in section 4.1. Attacks against the cluster's private sector have been the subject of previous research. We defend the cluster private segment, and we  also protect the global bag.

  • The anonymization technique in 4.1 must be used to accomplish our suggested strategy. We must first develop a global bag since we work on global bag security. We first separate private and non-private segments in the following ways, and then the private segment is further split into cluster private and global private segments. We can protect ourselves against composition, transitive composition, and our newly discovered critical absence and addition concerns in the global bag by doing so. After completing these steps, we are still vulnerable to composition and transitive composition attacks, which may be mitigated by using perturbation in the cluster private segment. Because of the dynamic nature of data, we're also dealing with global bag privacy, and even after providing a distinct global bag idea, it's still vulnerable to assaults. Furthermore, there is still a link between the private and non-private segments even after segregation. So, in section 4.3, we'll show how to improve the security of serial data. In response to your suggestions, in section 4.3 we made every effort to enhance our writing style in order to better communicate the creative nature of our approach and present it in a more thorough way.
  • It is suggested to present the privacy analysis of the method in a separate subsection. Also, the conclusion section is too verbose, just focusing on expressing the contribution of the work and the results obtained.
  • We will take into account the valuable comments you made while reviewing our manuscript. We are delighted to improve the privacy analysis section in accordance with your helpful suggestions. The above-mentioned enhancement has been implemented in the relevant section.
  • The organization and language need to be revised for readability. The article does not express the author's work concisely. There is too much superfluous content. In addition, attention should be paid to the grammar and formatting of the article. (For example, page 6, line 11, "is know" should be revised to "is known", page 6, line 13 "At time n ≤1", missing space. "Page 7, Definition 8, "Let t* belongs" should be revised to "Let t* belong", etc.)
  • We eliminate any redundant text from the revised document and make every effort to convey our work in a more concise manner. All grammatical and formatting errors have been corrected in this text. We sincerely regret any inconvenience this has faced by you. All of the previously mentioned errors have been rectified as well. Once again thanks on your remarks.

Reviewer 2 Report

Overall, the paper addresses a very important, bud widely explored issue of data privacy. The manuscript is clear, relevant for the field and presented in a well-structured manner. However, there are a few places that need a revision or elaboration:

  1. The keywords section should be developed.
  2. Tables 1,2,3, should have unique titles (release 1, release 2…).
  3. Each symbol presented in the table e.g. Ts, Tg should be precisely defined, explained (Tables 4,5,6) before it was used.
  4. The labels in the figures are hard to read. Please increase the font size (e.g. fig.1, fig. 2, fig.4 …).
  5. Please correct some editorial errors like timen <= 1 (line 208), data breach.For (line 229)
  6. For better readability please add the explanation to the symbols used in Figure 3. (near the equation). Fig.3 should be formatted, numbered like equation, rather than a figure.
  7. Please provide more information on the datasets used for the test.
  8. I advise to put two charts (Figure 8) in to one chart with two series. Presented results will be easier to compare (one figure, same scale). The same with figure 9.
  9. Please use the same scale for Perturbation Rate in figure 10.
  10. Add the numbering to equation presented in line 478.
  11. The labels in figure 11. Should be moved outside the columns.
  12. Please explain why we can observe the breaking point on the graph (sample size 5%) and why for 0% sample size the runtime is about 600s (fig. 12).
  13. As far as the methodological chapter and the research procedure have been described in detail, the Experimental Evaluation paragraph is rather short. Please develop the section addressing mostly the differences and breaking points in the charts.
  14. I’m not sure if the authors should use the statement “kanony an ideal strategy”, line 508
  15. The authors should also address how the proposed method can be adapted to other fields. Please add the information about the possibility of using of developed method in other fields for different types of serial data.
  16. Do the research/methods have any limitations, if so, please indicate it.
  17. The cited references were published mostly more than ten years ago. Please add more up to date literature (published within the last 5 years).

Pros:

  • well throughout study and well written manuscript,
  • description of algorithm that maximizes the data’s usefulness and provides a stronger privacy guarantee,
  • clear research objectives and focus of the study,
  • systematic experimental analysis illustrating the theoretical framework.

Cons:

  • short results and discussion paragraph.

Author Response

A Novel Privacy Paradigm for Improving Serial Data Privacy

Ayesha Shaukat, Adeel Anjum, Saif U. R. Malik, Munam Ali Shah, Carsten Maple

We would like to thank the reviewer for the valuable feedback and suggestions. We believe that incorporating these comments have significantly improved the quality of the work. Following is the point-by-point response to the comments.

Reviewer 2

Comments and Suggestions for Authors

Overall, the paper addresses a very important, bud widely explored issue of data privacy. The manuscript is clear, relevant for the field and presented in a well-structured manner. However, there are a few places that need a revision or elaboration.

Response:

  • Thank you for taking the time to provide such valuable comments. We appreciate you taking the time to read our manuscript and provide feedback. We take into consideration the comments and suggestions of reviewers. The primary goal of the proposed effort is to protect the privacy of serial data while it is being shared and used widely today in health records, bank records etc.
  • The keywords section should be developed.
  • To comply with the comments and suggestions, this keyword part has been revised. We appreciate you taking the time to write reviews for us. We will make every effort to update our document in a well-structured, thorough way in response to your suggestions. Thank you for your time.

  • Tables 1,2,3, should have unique titles (release 1, release 2…).
  • We were missing a correct explanation of the tables, which would have allowed us to create unique names that would have helped us grasp the progression of the content. We sincerely regret for this error, and we have corrected it in the amended version by using distinct titles.

  • Each symbol presented in the table e.g. Ts, Tg should be precisely defined, explained (Tables 4,5,6) before it was used.
  • In accordance with your reviews, the symbols table will be placed before the tables presenting the convicts' health record. Thank you very much for your assistance in improving our manuscript. All of the updated work will be included in the revised script.
  • The labels in the figures are hard to read. Please increase the font size (e.g. fig.1, fig. 2, fig.4 …).
  • I'd want to thank you again for your comments and critiques, which have all contributed to the improvement of our manuscript. Each and every one of the reviewer's criticisms and recommendations has been taken into consideration and fully incorporated into the revised text. As a result of raising font size, figures deform in their shape. Therefore we increase figure size and as a result of increasing figure size, we hope you will not have any difficulty in reading any longer.

  • Please correct some editorial errors like timen <= 1 (line 208), data breach.For (line 229).
  • We have corrected all typo,formatting and grammatical errors as per reviewers suggestions.
  • For better readability please add the explanation to the symbols used in Figure 3. (near the equation). Fig.3 should be formatted, numbered like equation, rather than a figure.
  • We are sorry on our mistake of writing equation in wrong style. We have corrected in revised version like this.

  • It is now in equation tag. Moreover, we have added symbols detail near equation in manuscript in section 4.2 although it is already in table 1.
  • Please provide more information on the datasets used for the test.

In the manuscript we have added more details about datasets in experimental section.  These transactions are actually click streams from two online retailers. We have elaborated description about no.of transactions, max trans. Length etc in manuscript and add more details about sample sizes and timestamps. We have also make table 11 for clear understanding about datasets.

  • I advise to put two charts (Figure 8) in to one chart with two series. Presented results will be easier to compare (one figure, same scale). The same with figure 9.

We have put two charts into one as per your instructions. Moreover we have also elaborated all the graphs discussion in experimental section.

  • Please use the same scale for Perturbation Rate in figure 10.
  • Thank you very much for your thoughts and evaluations, which have helped to enhance the paper. In the updated paper, all of the reviewer's criticisms and recommendations have been fully implemented. We have now use the same scale to calculate perturbation rate as in figure 8 and 9.

  • Add the numbering to equation presented in line 478.
  • It was editorial error thanks for mentioning this issue. We have improved in updated manuscript.
  • The labels in figure 11. Should be moved outside the columns.
  • We have move labels at the bottom. Now it is easily observailable by looking tops of bars. Moving outside create difficulty in adjusting two graphs adjacently.
  • Please explain why we can observe the breaking point on the graph (sample size 5%) and why for 0% sample size the runtime is about 600s (fig. 12).
  • We are really grateful to you for devoting your valuable time to our paper, and we are delighted by your response. We take several breaking points on the graph for calculating runtime, such as 1.25 percent, 5 percent, and 8.75 percent of sample sizes, and we computed the time required by our technique vs the time required by sanony's approach. Because we are interested in comparison graphs of perturbation rate, error rate, and time, we are free to use sample sizes as taken in the prior method to dealing with difficulties arising in tackling serial data. We have the option of changing the sample sizes, but we prefer to stick with the ones we have for appropriate comparisons. Furthermore, the time is not 600s at sample size 0 percent; rather, it is the time calculated at sample size 1.25 percent. We have  recognised that the graph does not convey its meaning properly, and so we have replaced these graphs to ensure correct comprehension. Once again, thank you for your help and advice.
  • As far as the methodological chapter and the research procedure have been described in detail, the Experimental Evaluation paragraph is rather short. Please develop the section addressing mostly the differences and breaking points in the charts.
  • Thank you for giving your valuable time. It really means to us. We provide more information in all the experimental evaluation parageaphs and improve the graphs as per your guidelines. We have also tried to clearly presented differences of our approach with sanony and mention breaking points separately in explanation of all experiments. You can now easily understand experimental evaluation paragraphs in experiment’s section.
  • I’m not sure if the authors should use the statement “kanony an ideal strategy”, line 508
  • Despite the fact that we provided an extremely safe and privacy-preserving architecture in which all conceivable sites of data leakage were taken into consideration, it is a truth that we will never be able to attain an optimal stage in terms of privacy preservation. We sincerely apologise for any inconvenience this has caused you, and we have made the necessary corrections. Thank you for taking the time to read over everything.

  • The authors should also address how the proposed method can be adapted to other fields. Please add the information about the possibility of using of developed method in other fields for different types of serial data.

  • Using a prisoner's health record as an example, we pay close attention to and spot critical attacks in serial data. As we all know, serial data is dynamic, which is why these attacks are possible, and we live in an era when dynamic data is prevalent. Our technique is quite broad in scope and may be used in various forms of serial data. There are various sorts of data that we may utilize the techniques described in this article for, such as bank records where there are multiple sensitive values of an individual like its salary performing two jobs at the same time, bank account holder having two accounts at the same time, moreover in personal profiles, and business data.

  • Do the research/methods have any limitations, if so, please indicate it.
  • The drawback we've found with this strategy is that it causes a tiny increase in relative error, but it also improves privacy compared to earlier approaches. Further while giving protection to multiple sensitive values in serial data we have found some future work which we will mention in manuscript.
  • The cited references were published mostly more than ten years ago. Please add more up to date literature (published within the last 5 years).
  • We have included the most recent articles in our literature; but, in order to get a full understanding of our approach, it is necessary to include some older studies as well, in order to explain certain words and so on. Our method is useful when dealing with data that contains one or more private keywords. In comparison to other ways, it protects the privacy of serial data more effectively and efficiently. Our system is advantageous in that it guards against composition, transitive composition, critical absence concerns, and correlation attacks, among other things. We have added articles in accordance with your directions in the literature review section. These articles are listed below.
  • Anjum, Adeel, et al. "An efficient approach for publishing microdata for multiple sensitive attributes." The Journal of Supercomputing 74.10 (2018): 5127-5155.

  • Khan, Razaullah, et al. "Privacy preserving for multiple sensitive attributes against fingerprint correlation attack satisfying c-diversity." Wireless Communications and Mobile Computing 2020 (2020).

  • Sajjad, Haider, et al. "An efficient privacy preserving protocol for dynamic continuous data collection." computers & security 86 (2019): 358-371.
  • Beg, Saira, et al. "A privacy-preserving protocol for continuous and dynamic data collection in IoT enabled mobile app recommendation system (MARS)." Journal of Network and Computer Applications 174 (2021): 102874.

  • Kanwal, Tehsin, et al. "A robust privacy preserving approach for electronic health records using multiple dataset with multiple sensitive attributes." Computers & Security 105 (2021): 102224.

Pros:

well throughout study and well written manuscript,

description of algorithm that maximizes the data’s usefulness and provides a stronger privacy guarantee,

clear research objectives and focus of the study,

systematic experimental analysis illustrating the theoretical framework.

Cons:

short results and discussion paragraph.

Round 2

Reviewer 1 Report

In the current version, more details have been added to highlight the contribution of the work in this paper. Although some of the content was modified in a different way than expected (e.g., Tables 1-3, the authors clearly conveyed the role of the tables by adding "Release" tags). The work in this article complements the work in reference 9 by building on the "global bag" approach, focusing on the security risks associated with missing keywords in the “global bag” and the continuous addition of data in subsequent versions. The proposed approach is complete, addresses the issues raised in the introduction, and provides a security analysis of the scheme with sufficient experimental content.

A few suggestions and comments for improvement:

1. The “Literature Review” subsection can be divided into several paragraphs, which will more clearly indicate the development process of this type of work.

2. In section 4.2, the introduction of the basic concepts of “Backward perturbation” and “Forward perturbation” can be added appropriately.

3. The authors have checked the grammar and formatting of the entire article and have reduced many errors compared to the previous version. However, there are still a few unnecessary errors, for example, on page7, line 16 ".A" is missing spaces.

Author Response

A Novel Privacy Paradigm for Improving Serial Data Privacy

Ayesha Shaukat, Adeel Anjum, Saif U. R. Malik, Munam Ali Shah, Carsten Maple

Round 2

03-03-2022

Comments and Suggestions for Authors

In the current version, more details have been added to highlight the contribution of the work in this paper. Although some of the content was modified in a different way than expected (e.g., Tables 1-3, the authors clearly conveyed the role of the tables by adding "Release" tags). The work in this article complements the work in reference 9 by building on the "global bag" approach, focusing on the security risks associated with missing keywords in the “global bag” and the continuous addition of data in subsequent versions. The proposed approach is complete, addresses the issues raised in the introduction, and provides a security analysis of the scheme with sufficient experimental content.

A few suggestions and comments for improvement:

  1. The “Literature Review” subsection can be divided into several paragraphs, which will more clearly indicate the development process of this type of work.

We are delighted to see such a great reaction to our request. As you correctly point out, having multiple paragraphs in the literature portion that are more readily accessible and convey their idea effectively is beneficial.

  1. In section 4.2, the introduction of the basic concepts of “Backward perturbation” and “Forward perturbation” can be added appropriately.

We are thankful for your comments. We try to elaborate the concepts of BP and FP for clear understanding. We make two subsegments of the private segment into the global private segment and cluster private segment. We are working with serial data in which the records are continuously added and deleted. Various attacks arise after segmentation which includes composition and transitive composition attacks.  To overcome these attacks, backward and forward perturbation is introduced, which involve the use of counterfeits in cluster private segment. Backward perturbation adds counterfeits in the cluster private segment to reduce "composition assaults" resulting from transactions being connected to a previously disclosed corpus. Because of these counterfeits, overlap reduces by altering cover Ω(Co), a group of overlaps between two corpora. Through backward perturbation, overlapping with previously published corpora can be reduced. Next, FP is applied on derived clusters by further adding counterfeits due to transitive composition attacks. Derived clusters are clusters formed after the addition and deletion of records in subsequent releases. We are aware that records are constantly being added and deleted, resulting in further counterfeits to the cluster private segment to be released after BP has ensured the security of derived clusters. After this, all transactions remain serially preserved from composition and transitive composition attacks and we further focus on correlation and critical absence effect in global bag.

  1. The authors have checked the grammar and formatting of the entire article and have reduced many errors compared to the previous version. However, there are still a few unnecessary errors, for example, on page7, line 16 ".A" is missing spaces

Thank you so much for taking the time to respond. We sincerely apologies for any inconvenience this has caused you. We are working to remedy the issue and hope that you will not notice it in the new version.
